# Synthesis and Characterization of α-Al_2_O_3_/Ba-β-Al_2_O_3_ Spheres for Cadmium Ions Removal from Aqueous Solutions

**DOI:** 10.3390/ma15196809

**Published:** 2022-09-30

**Authors:** Pamela Nair Silva-Holguín, Álvaro de Jesús Ruíz-Baltazar, Nahum Andrés Medellín-Castillo, Gladis Judith Labrada-Delgado, Simón Yobanny Reyes-López

**Affiliations:** 1Departamento de Ciencias Químico-Biológicas, Instituto de Ciencias Biomédicas, Universidad Autónoma de Ciudad Juárez, Envolvente del PRONAF y Estocolmo s/n, Ciudad Juárez 32300, Mexico; 2CONACYT-Centro de Física Aplicada y Tecnología Avanzada, Universidad Nacional Autónoma de México, Boulevard Juriquilla 3001, Santiago de Querétaro 76230, Mexico; 3Centro de Investigación y Estudios de Posgrado, Facultad de Ingeniería, Universidad Autónoma de San Luis Potosí, Av. Dr. Manuel Nava No. 8, San Luis Potosí 78210, Mexico; 4Instituto Potosino de Investigación Científica y Tecnológica, San Luis Potosí 78216, Mexico

**Keywords:** α-Al_2_O_3_/Ba-β-Al_2_O_3_, spheres, cadmium, adsorption, phase transformation

## Abstract

The search for adsorbent materials with a certain chemical inertness, mechanical resistance, and high adsorption capacity, as is the case with alumina, is carried out with structural or surface modifications with the addition of additives or metallic salts. This research shows the synthesis, characterization, phase evolution and Cd(II) adsorbent capacity of α-Al_2_O_3_/Ba-β-Al_2_O_3_ spheres obtained from α-Al_2_O_3_ nanopowders by the ion encapsulation method. The formation of the Ba-β-Al_2_O_3_ phase is manifested at 1500 °C according to the infrared spectrum by the appearance of bands corresponding to AlO_4_ bonds and the appearance of peaks corresponding to Ba-O bonds in Raman spectroscopy. XRD determined the presence of BaO·Al_2_O_3_ at 1000 °C and the formation of Ba-β-Al_2_O_3_ at 1600 °C. Scanning electron microscopy revealed the presence of spherical grains corresponding to α-Al_2_O_3_ and hexagonal plates corresponding to β-Al_2_O_3_ in the spheres treated at 1600 °C. The spheres obtained have dimensions of 4.65 ± 0.30 mm in diameter, weight of 43 ± 2 mg and a surface area of 0.66 m^2^/g. According to the curve of pH vs. zeta potential, the spheres have an acid character and a negative surface charge of −30 mV at pH 5. Through adsorption studies, an adsorbent capacity of Cd(II) of 59.97 mg/g (87 ppm Cd(II)) was determined at pH 5, and the data were fitted to the pseudo first order, pseudo second order and Freundlich models, with correlation factors of 0.993, 0.987 and 0.998, respectively.

## 1. Introduction

Alumina is a chemical compound formed by oxygen and aluminum whose molecular formula is Al_2_O_3_ (aluminum oxide). It is the most widely used oxide-based ceramic, either in its pure form or as a raw material for mixtures with other oxides. Aluminum oxide, also known as alumina, is one of the most important oxides used industrially; it has a high melting temperature of 2054 °C and is chemically very stable and non-reactive, leading to applications such as high-temperature components and biomedical implants. Alumina can be found in several metastable transition phases determined by the temperature and precursor employed to obtain the oxide. The alumina phases are generally referred to as transition alumina (due to their metastable nature compared to α-alumina). They are classified into two groups: one is a low-temperature transition alumina, which dehydrates below 600 °C (rho (ρ), chi (χ), eta (η) and gamma (γ)-alumina); and two is a high-temperature transition alumina obtained between 900 and 1000 °C (kappa (κ), delta (δ) and theta (θ) alumina). The principal phase transition in alumina follows the sequence γ-Al_2_O_3_ → δ-Al_2_O_3_ → θ-Al_2_O_3_ → α-Al_2_O_3_; the sequence of transition of alumina that is formed depends strongly on the starting raw material [1]. Each of the phases presents different characteristics properties that allow it to be used for diverse applications such as reinforcement [2], catalysts [3], refractories [4], and adsorbents [5]. The most stable and widely utilized alumina phase is the alpha phase. α-Al_2_O_3_ is used as an advanced ceramic given its high chemical stability, hardness, high refractoriness, high thermal conductivity, good wear resistance, and electrical insulation. Some applications include the fabrication of reinforcers, electronics, photonics, sensors, and catalysts. The electrical resistance of alumina is high, it has excellent optical transparency, and together with additives such as chromium and titanium, it is important as a gemstone (sapphires and rubies) [6,7].

In recent years, more and more attention has been paid to the development of alumina-based nanostructured ceramic materials for advanced applications such as catalysts, electronics, metals, and alloy manufacturing, as well as separation and purification. γ-alumina is an extremely important phase that has been used as a catalyst and catalyst substrate in the automotive and petroleum industries, structural compounds for spacecraft, abrasive and thermal wear coatings, and wastewater separation and purification. Recently, it has been established that γ-alumina is thermodynamically stable relative to α-alumina when a critical surface area is achieved. Several conventional processes have been developed to synthesize alumina nanostructures or combinations of alumina as in the β-Al_2_O_3_ phase; the processes include mechanical synthesis [8], vapor phase process [9], precipitation [10], microwave [11] and sol–gel combustion [12].

The β-Al_2_O_3_ consists of blocks of spinel (tightly packed layers) and conduction layers (loosely packed layers) that are stacked alternately to form a layered structure with plate morphology. Spinel blocks are composed solely of Al^3+^ and O^2−^ ions, with Al^3+^ ions occupying the octahedral and tetrahedral interstices in spinel. The spinel blocks are separated by a conduction layer where the incorporated cations (Na^+^, K^+^, Ba^+2^, etc.) are housed and by oxygen ions bound to the neighboring spinel blocks through two aluminum ions, conducting to a hexagonal structure, even though the spinel is cubic [13,14,15].

Barium aluminates have several types of stoichiometric compositions, from mono-aluminate to barium hexa-aluminate. Barium hexa-aluminate has attracted a lot of attention in some potential applications, including catalytic combustion and gas sensing, because of its high chemical, physical, and thermal stability as well as high ionic conductivity and high resistance to sintering and thermal shocks. In addition, barium hexa-aluminate can function as a reinforcement for ceramic matrix composites. The existence of the hexa-aluminate phase with a layered structure can improve the fracture toughness of Al_2_O_3_ matrix compounds. In addition to the stoichiometric aluminates, barium can form many non-stoichiometric Ba-O-Al complex aluminates. There are many methods to synthesize barium hexa-aluminate, such as solid-state reaction, sol–gel process, hydrothermal precipitation, alkoxide route and inductively coupled plasma spray techniques. Most of these methods require additional high-temperature heat treatment before forming the barium hexa-aluminate phase [8]. In addition, the chemical or physical properties of alumina materials can be exploited for efficient storage materials and constitute an important subject in material research. For this reason, the synthesis of α-Al_2_O_3_ and β-Al_2_O_3_ ceramics materials at low temperatures has been studied. The characterization of thermal decomposition behavior can demonstrate the phase transformation process and help determine the optimum conditions for the process and the temperatures where the phases appear [16]. The present investigation shows the synthesis of α- Al_2_O_3_/Ba-β-Al_2_O_3_ spheres starting from α-Al_2_O_3_ nanopowders by the ionic encapsulation method.

In this sense, is important to mention that the aim and scientific relevance of this work is focused on the development of new ceramic materials with high remotion efficiency, which were studied and evaluated from the allotropic phases of Al_2_O_3_ obtained during the synthesis process. The allotropic phases such as α-Al_2_O_3_/Ba-β-Al_2_O_3_ and Ba-β-Al_2_O_3_ have not been reported from the adsorption of heavy metal point of view. In this work, detailed kinetic adsorption models and potential are presented. The results obtained describe and discuss the Cd adsorption process and the high remotion efficiencies exhibited by the ceramic-metallic composites.

## 2. Materials and Methods

The α-Al_2_O_3_/Ba-β-Al_2_O_3_ spheres were obtained by the encapsulation method by ion exchange using sodium alginate as the encapsulating agent. A mixture of the base material of 10–30 nm nanometric powders of α-Al_2_O_3_ (original α-Al_2_O_3_), water, sodium alginate and PVA was made in proportions of 55, 30, 10 and 5%, respectively. The mixture was dropped dropwise into a 0.6 M barium chloride solution where the ion exchange of monovalent Na^+^ ions for divalent Ba^+2^ ions takes place in the alginate structure according to Reaction (1), giving rise to the formation of semi-rigid spherical bodies upon contact with barium chloride [17,18].
(R1)$${\mathrm{BaCl}}_{2}+2\left(C_{6}H_{7}O_{6}-\mathrm{Na}\right)\to 2{\mathrm{Na}}^{+}+2{\mathrm{Cl}}^{-}+C_{6}H_{7}O_{6}-\mathrm{Ba}-C_{6}H_{7}O_{6}$$

The spheres are left to settle in the barium chloride solution for 24 h to allow further diffusion of Ba^+2^ into the sphere. After that time, they were given a heat treatment at 100 °C to eliminate the water.

Phase evolution and microstructure were determined by FTIR infrared spectroscopy, Raman spectroscopy, X-ray diffraction and scanning electron microscopy. FTIR was carried out using a Bruker Alpha Platinum ATR spectrometer, obtaining 48 scans per sample with a resolution of 4 cm^−1^ in a spectral range from 4000 to 400 cm^−1^. Using a Raman Confocal alpha 300 WiTec spectrometer, an integration time of 0.5 s, 10 accumulations and a 532 nm excitation laser source, the characterization of the composites was carried out. The composites were placed over carbon tape on a slide for its microstructural analysis, which was made in a Field Emission Scanning Electron microscope SU5000 Hitachi with an energy of 20 keV, and an Energy-Dispersive X-ray Spectroscopy (EDS) assay was carried out to know the elemental composition. Spheres were analyzed by the powder diffraction technique, in instrument X’Pert PRO PANalytical, with Cu kα = 1.54, 20 kV, in the range of 2θ of 10 to 80° and a 2°/min scanning speed. The textural properties (specific area, pore volume and pore size) were determined by N_2_ physisorption using a Micromeritics equipment, model ASAP 2010 and using the Brunauer–Emmett–Teller (BET) method. The zeta potential (ζ) was determined in a particle analyzer (HORIBA, SZ-100) at different pH. The phase evolution was followed in two forms: first, the green spheres were ground and powder heat treated at 100, 600, 1000, 1400, 1500 and 1600 °C and designated as P-Al_2_O_3_ 100 °C, P-Al_2_O_3_ 600 °C, P-Al_2_O_3_ 1000 °C, P-Al_2_O_3_ 1400 °C, P-Al_2_O_3_ 1500 °C and P-Al_2_O_3_ 1600 °C, and second, the spheres were also subjected to a heat treatment of 1600 °C and designated as S-Al_2_O_3_ 1600 °C.

The adsorbent capacity of alumina spheres treated at 1600 °C was determined by kinetic and equilibrium studies of Cd(II) adsorption. A stock solution of 1000 mg/L of Cd(II) was prepared from cadmium nitrate tetrahydrate Cd(NO_3_)_2_·4H_2_O (Jalmek^®^), which was diluted to obtain different initial concentrations of Cd (II); the dilutions were made with a pH 5 buffer solution of HNO_3_ and NaOH. For the adsorption kinetic studies, four spheres were used in 100 mL of 200 mg/L Cd^+2^ solution, keeping the pH constant at 5 by adding 0.1 M NaOH or HNO_3_ at room temperature. The equilibrium data obtained from the adsorption experiments were fitted on Pseudo first order (1), Pseudo second order (2), Elovich (3) and Intraparticular diffusion (4) kinetics models [19,20].
(1)$$q_{t}=q_{e}\left(1-e^{-\left(k_{1}t\right)}\right)$$
(2)$$q_{t}=\frac{k_{2}{q_{e}}^{2}\;t}{1+k_{2}q_{e}\;t\;}$$
(3)$$q_{t}=\beta In\left(\alpha \beta t\right)$$
(4)$$q_{t}=K_{DI}t^{1/2}$$
where *q_t_* is the amount of compound adsorbed over time (mg/g), *q_e_* is the amount of compound adsorbed at equilibrium (mg/g), *K*_1_ is the pseudo first order rate constant of the adsorption process (min^−1^), *K*_2_ is the pseudo second order constant (g mg^−1^ h^−1^), *α* is the sorption constant of the compound (mg g^−1^), *β* is the desorption constant (mg^−1^ g), *K_DI_* is the intraparticle diffusion rate constant (mg/g/min) and *t* is the time.

For adsorption equilibrium studies, dilutions were made at 20, 50, 100, 200, 400, 600 and 800 mg/L of Cd^+2^; a sphere was placed in 25 mL of each dilution and left in contact for 3 days at pH 5 and room temperature. Then, the equilibrium data obtained from the adsorption experiments were fitted using Langmuir (5), Freundlich (6) and Temkin (7) isotherm models [19,20].
(5)$$q_{e}=\frac{q_{m}bC_{e}}{1+bC_{e}}$$
(6)$$q_{e}=K_{F}{C_{e}}^{\frac{1}{n}}$$
(7)$$q_{e}=\frac{RT}{b_{T}}In\;\left(K_{T}C_{e}\right)$$
where *C_e_* is the concentration at equilibrium (mg/L), *q_m_* is the monolayer adsorbent capacity (mg/g), *b* is the Langmuir constant related to free adsorption energy (L/mg), *K_F_* is the Freundlich constant, which indicates the adsorbent capacity (mg/g), 1/n is the adsorption intensity, *R* is the gas constant (8.314 J/mol K), *T* is the temperature (K), *B_T_* is the constant related to adsorption temperature (L/g), and *K_T_* is Temkin’s constant.

The determination of the cadmium concentration was carried out in a selective ion equipment (ISE) and a Thermo scientific Dual star pH/ISE Benchtop electrode for cadmium. The adsorption capacity was determined by means of a mass balance represented by the following Equation (8):(8)$$q=\frac{\left(V_{i}*C_{i}\right)-\left(V_{f}*C_{f}\right)}{m}$$
where *V_i_* is the initial volume (L), *C_i_* is the initial concentration (mg/L), *V_f_* is the final volume (L), *C_f_* is the final concentration (mg/L), and *m* is the adsorbent mass (g).

Finally, the correlation factors *R*^2^ were calculated by the following mathematical expression:(9)$$R^{2}=\frac{\sum {\left(q_{m}-{\bar{q}}_{t}\right)}^{2}}{\sum {\left(q_{m}-{\bar{q}}_{t}\right)}^{2}+\sum {\left(q_{m}-{\bar{q}}_{t}\right)}^{2}}$$
where *q_m_* is the amount of Cd adsorbed by the ceramic–metallic composites at any time, *t*, expressed in mg/g (theoretical value). Analogous, *q_t_*, is the amount of Cd adsorbed by the absorbents at any time, *t* (mg/g), obtained from experimental data and ${\bar{q}}_{t}$ is the average of *q_t_* (mg/g).

## 3. Results and Discussion

The ceramic spheres at 1600 °C obtained have dimensions of 4.65 ± 0.30 mm in diameter, with a sphericity of 0.94 ± 0.07 and a weight of 43 ± 2 mg. The spheres without thermal treatment are observed in Figure 1A, with thermal treatment at 1600 °C in Figure 1B and for the dimension size of the spheres, several spheres are shown in a petri dish (Figure 1C).

Figure 2 shows the infrared spectra of the alumina powders and spheres treated at different temperatures. α-Alumina was used as base material in obtaining the spheres; the spectrum of the original α-alumina shows three bands at 631, 555 and 489 cm^−1^ corresponding to the vibrational modes AlO_6_ octahedral coordination denoting the crystalline structure of corundum. Thermal treatment below 1400 °C did not give a structural change in the α-alumina, as can be seen in Figure 2A. At 1500 °C, a structural change is observed with the appearance of a strong and wide absorption in the region of 937 to 582 cm^−1^ corresponding to vibrational modes AlO_4_ tetrahedral coordination, denoting the formation of an alumina of type spinel [21,22]. Comparing the powdered sample P-Al_2_O_3_ 1600 °C and sphere S-Al_2_O_3_ 1600 °C as observed in spectrum 2B, the powder thermally treated sample shows higher intensity in the bands corresponding to AlO_4_ vibrational modes compared to the material in spherical shape. The form in which the powder or sphere material is when subjected to the treatment determines the intensity of the bands corresponding to the phase change. In the infrared spectra, the presence of bands corresponding to water within the structure is not observed. The spectra treated at 1500 and 1600 °C (powder and sphere) show a discrete and weak band located at 1050 cm^−1^ corresponding to the surface vibration modes of tense Al-O bonds located in the surface layer related by the dihydroxylation of the alumina, which is characteristic of highly dehydrated alumina [23]. When the original α-Al_2_O_3_ is subjected without undergoing ionic encapsulation to 1600 °C, the appearance of the band at 1050 cm^−1^ corresponding to dehydration is not observed, so this band is characteristic of the dehydration of the spinel formed.

Figure 3 shows the Raman spectra of the powders and alumina sphere treated at different temperatures. In the case of the heat treatment at 1600 °C (powder and sphere), the appearance of bands below 1500 cm^−1^ is observed, which increase in intensity as they are found in powder form as opposed to being in a sphere. As the material is spherical, the phase change is more pronounced on the surface, leaving the interior unreacted, so the phase transformation is more pronounced in the powder. The peaks located at 136 and 279 cm^−1^ correspond to the Ba–O bond vibrations, while the peaks located at 389, 416, 576, 645 and 750 correspond to the AlO_4_ vibrations [24].

The XRD patterns in Figure 4 show the phase evolution of the alumina powders heat treated at 600, 1000 and 1600 °C. The alumina used as a base material for the synthesis of the spheres shows characteristic crystal planes of α-alumina (0 1 2), (1 0 4), (1 1 0), (1 1 3), (0 2 4), (1 1 6), (0 1 8), (2 1 4) and (3 0 0) located at 25°, 35°, 37.5°, 43°, 52°, 57°, 61.5°, 66° and 68°, respectively, according to JCPDS file no. 83-2081 [25]. Up to 600 °C, the XRD pattern shows the presence of α-alumina (92.67%) and γ-Al_2_O_3_ (7.21%) with crystal planes (2 2 2), (4 0 0) and (4 0 0) located at 39°, 46° and 67°, respectively. When the temperature is increased to 1000 °C, peaks corresponding to a barium aluminate BaO∙Al_2_O_3_ (2.45%) were detected, being the most intense peak at 2θ = 28.35° (hexagonal lattice $\left(3\bar{12}0\right)$). At 1600 °C, the quantification of barium aluminate decreases to 0.84%, which suggests that BaO∙Al_2_O_3_ is an intermediate phase [8]. At 1600 °C, the presence of γ-Al_2_O_3_ and the formation of spinel Ba-β-Al_2_O_3_ are observed, whose mean diffraction peaks are located at 19.01° ($10\bar{1}1)$, 26.73° ($10\bar{1}3)$ and 31.23° ($2\bar{11}0)$.

The synthesis mechanism of spinel Ba-β-Al_2_O_3_ in the spheres begins first with the diffusion of barium throughout the sphere substituting sodium in the alginate structure according to Reaction (1). The barium is complexed with the alginate; as the heat treatment is given, increasing the temperature, the organic matter is lost and transformed into CO_2_ and water, releasing the barium from the alginate, and the formation of barium oxide occurs. At 600 °C, an increase in the crystalline phase of gamma alumina is observed with the decrease in the percentage of the α-Al_2_O_3_ base material; therefore, the presence of barium decreases and breaks the network of the alpha alumina structure with the formation of the gamma alumina phase. At 1000 °C, the gamma alumina with the barium oxide gives the formation of the BaO·Al_2_O_3_ phase mainly on the surface because it reacts more easily with oxygen, while the gamma alumina found inside the sphere transforms back to the alpha alumina phase. Another phase formed in a smaller proportion is Ba-β-Al_2_O_3_. The process occurs first on the surface and from there toward the core of the sphere. Due to a higher percentage of BaO∙Al_2_O_3_ on the surface of the sphere, it is indicative that the barium found in the core of the sphere diffuses to the surface due to the oxidation concentration gradient due to the Kirkendall effect [26]. Finally, at 1600 °C, the rearrangement of the BaO∙Al_2_O_3_ phases occurs, which allows the increase in the formation of Ba-β-Al_2_O_3_ and γ-Al_2_O_3_. The formation of beta alumina is temperature dependent; a certain amount of energy is necessary for the structural rearrangement to occur. The observed phase transformations with respect to temperature can be followed according to the following labeled reactions in Figure 5.

Figure 6 shows the XRD patterns for the powder and the alumina sphere treated at 1600 °C. In the two samples analyzed, we find the same crystallographic planes for the α, γ and β alumina; however, the phase quantification is different. As shown in Table 1, the difference lies in a lower percentage of formation of the γ-Al_2_O_3_, BaO∙Al_2_O_3_ and Ba-β-Al_2_O_3_ phases in the sphere, according to the results of infrared and Raman spectroscopy. When submitting a material in powder form to a heat treatment presents greater diffusion and reactivity, the decomposition of organic matter is uniform, as is its oxidation, so there is no Kirkendall effect such as that detected on the surface of the spheres. The percentage of beta alumina in the sphere is proportional to the percentage of sodium alginate used, for a higher proportion of beta alumina in the sphere, a higher proportion of sodium alginate is necessary in the initial mixture. Figure 7 shows the structures and network parameters of the structures found in the studied system.

SEM micrographs of the interior (core) and surface of alumina spheres treated at 1000 and 1600 °C are shown in Figure 8. Both on the surface and inside the sphere treated at 1000 °C, no defined morphology is observed in the analyzed area; particles of 250 ± 76 nm are observed in the form of thick agglomerates. When the sphere is treated at 1600 °C, the grain size grows and two morphologies with different densities can be seen according to the contrast in backscattered electrons. The particles with spherical morphology and opaque contrast correspond to the majority α-alumina phase with sizes of 1.17 ± 0.33 μm, while the particles with elongated morphology in the form of hexagonal plates and bright contrast correspond to the minority Ba-β-Al_2_O_3_ phase with length variables of 0.47 ± 0.08 μm. The surface and the interior of the sphere treated at 1600 °C present the same contrast but different morphology. This agrees with the proposed phase formation mechanism, where the formation of beta alumina in the core of the sphere is a slower and non-uniform process due to the decomposition of the organic matter from the surface toward the center of the sphere. Furthermore, oxidation is not uniform in the sphere, so the formation of the Ba-β-Al_2_O_3_ phase occurs first on the surface and later in the core.

Using the BET method, the textural properties of the alumina spheres were determined by nitrogen physisorption, and the adsorption isotherms were determined. Figure 9 shows the isotherms for the alumina spheres treated at 1000 °C (A) and 1600 °C (B); both isotherms have a form of isotherm type 3 according to IUPAC where the adsorbate–adsorbate interaction is greater compared to the adsorbate–sorbent interaction [27,28]. Table 2 shows the textural properties of the spheres at 1000 °C and 1600 °C, the specific area of the sphere decreases from 2.23 to 0.66 m^2^/g with the heat treatment. The sintering process consists of the consolidation of powders at high temperatures close to the melting temperature, where the material particles become a solid compact body, causing changes in its mechanical and physical properties [29]. At 1600 °C, the sintering process took place, and important changes in morphology were produced. During this step, the growth of grains and the decrease in the porosity of the material were observed, as shown in Figure 8. Due to the sintering process, the surface area is reduced; however, this same process helps to consolidate the material and obtain a manipulable material with high hardness, which is applicable for obtaining filters.

When appreciating differences in the morphology of the particles on the surface and inside the sphere, an experiment was carried out to determine the observed effect according to the methodology used for the encapsulation of alumina spheres (Reyes–Silva methodology). The 1600 °C treated alumina spheres were further heat treated at 1600 °C in two forms: non-fragmented sphere and fragmented sphere, and the morphology of the sphere center was observed by SEM. In Figure 10, the center of the non-fragmented sphere has a higher composition of α-alumina grains and a lower composition of hexagonal plates corresponding to β-alumina. The micrographs of the center of the fragmented sphere show a greater composition of more elongated hexagonal plates and with the presence of porosity in the particles. To increase the formation of the Ba-β-Al_2_O_3_ phase, a longer heat treatment time is necessary to allow the formation of beta alumina in the nucleus of the sphere. A thermal treatment of greater time or temperature causes the Ba-β-Al_2_O_3_ phase of the surface to give rise to the migration and detachment of barium with the formation of pores found in the hexagonal planes of Ba-β-Al_2_O_3_, as shown in Figure 11 [11]. It is to be observed that the loss of Ba or BaO propitiates that in the spheres, only the corundum-type structure exists again.

Using a microelemental analysis by EDS, the elemental composition of the two morphologies present on the surface of the alumina spheres treated at 1600 °C was determined (Figure 12). The opaque morphology shows 57.35% aluminum, 40.54% oxygen and 2.11% barium, while the bright contrast morphology shows 51.49% aluminum, 39.85% oxygen and 8.66% barium, which confirms and indicates that barium reacts with α-alumina and forms β-alumina. Figure 13 shows a mapping of the elemental distribution of the surface of the spheres; in the micrograph, hexagonal planes corresponding to β-alumina can be observed, and through the mapping, it is observed that the grains have less presence of barium as is circled in red on the image.

Figure 14 shows the curves of zeta potential vs. pH for the materials P-Al_2_O_3_ 100 °C, P-Al_2_O_3_ 1000 °C, P-Al_2_O_3_ 1600 °C and S-Al_2_O_3_ 1600 °C. The P-Al_2_O_3_ 100 °C sample shows an isoelectric point (IEP) at pH 6.3 with a positive charge lower than this pH and a maximum positive zeta potential of +51.1 ± 2.3 mV at pH 3.84, while at higher pH, the surface charge is negative with maximum value at −37.3 ± 1.5 mV at pH 10.55, and the results agree with the IEP of α-alumina in deionized water at 6.7 [30]. The P-Al_2_O_3_ 1000 °C presents α-Al_2_O_3_ and BaO∙Al_2_O_3_ as phases, which causes a displacement of the IEP from 6.3 to 8, as shown in Figure 14B. In addition, an increase in the positive zeta potential can be observed up to +74.7 ± 1.4 mV at pH 5.2; similarly, the negative potential increases to −44.3 ± 2.6 mV at pH 10. The adsorption of divalent cations on the surface of the alumina causes the alteration of the surface charge of the alumina and a displacement of the IEP to the basics. Das et al. determined the variation of the zeta potential of alumina with the presence of divalent Ca^+2^ and Mg^+2^ cations separately at 5 × 10^−4^ mol/dm and observed an increase in the positive zeta potential at pH 5 of +22 mV to +25 mV, plus an IEP shift from 6.7 to 7.4. Figure 14C shows a change in the surface charge of the alumina treated at 1600 °C from positive to negative at acidic pH, while at basic pH, the negativity increases, crossing the isoelectric point at a pH lower than 2. The positive surface in alumina is due to the presence of protons on the surface at acidic pH; when treated at basic pH, the surface is negative due to deprotonation of the surface according to Reaction (2), where the surface of the material is represented by ≡. Heat treatment is a determining factor in the surface charge of alumina; the dehydration of alumina causes the movement of protons toward the surface which, together with the OH ions, forms water molecules that detach from the surface, leaving a negative surface [31]. The dehydration of alumina causes a negative zeta potential, which makes the material acidic and easily interacts with basic molecules through electrostatic attractions. The spheres have less exposed area, so the dehydration is not as pronounced as in the case of powder; this is observed in the magnitude of the change of the zeta potential toward the negative (Figure 14C,D).
(R2)$$\equiv \mathrm{Al}-{\mathrm{OH}}_{2}^{+}\;\begin{array}{c} \overset{{\;H}^{+}}{↔}\;\equiv \mathrm{Al}-\mathrm{OH}\;\overset{\;O-H^{-}\;}{↔}\;\equiv \mathrm{Al}-O^{-} \end{array}$$

According to the fact that the highest percentage of beta alumina is found on the surface of the sphere, an acid treatment of the material only causes a decrease in the negativity of the surface charge of the material without reaching neutrality at a pH greater than 2, as shown in Figure 14C, so the material can be reused by giving it an acid treatment to release the adsorbed heavy metal ions without causing the dissolution of the material.

The adsorbent capacity of Cd^+2^ on the alumina spheres treated at 1600 °C was evaluated, an adsorption kinetics and isotherm were performed, and the material was characterized after adsorbing cadmium by scanning electron microscopy and infrared spectroscopy to confirm the presence of cadmium on the material. Kinetic studies help determine the rate of adsorption and the time required for the process to reach equilibrium. Using the adsorption kinetics shown in Figure 15A, it was determined that a time of 2 days is sufficient to reach adsorption equilibrium. Four kinetic models were applied to describe the adsorption mechanism of Cd(II) on the alumina spheres treated at 1600 °C. Pseudo first order, pseudo second order, Elovich and intraparticle diffusion models were included, and the calculated kinetic parameters are shown in Table 3. To select the best fit of the models used, the average absolute percentage deviation (%D) of each model was calculated with the following equation [32]:(10)$$\%D=\left|\frac{q_{exp}-q_{cal}}{q_{exp}}\;\right|\times 100\%$$

According to the criterion of the lowest percentage of deviation, the pseudo first and pseudo second order models describe better the data. The pseudo first order model describes the adsorption rate and depends on the available sites in the adsorbent for the physisorption process. The pseudo second order model describes the adsorption reaction rate with energetically heterogeneous sites dependent on the adsorbent; it is considered a chemisorption model [33]. Figure 15B shows the adsorption isotherm; three models were applied to describe the adsorption of Cd(II) in the alumina spheres treated at 1600 °C. The models used were Langmuir, Freundlich and Temkin, and the kinetic parameters obtained are shown in Table 4. The model with the lowest percentage of deviation and the one that best describes the adsorption process was the Freundlich model. The Freundlich isotherm can describe the non-ideal reversible adsorption of multiple layers on a heterogeneous surface. K_F_ and 1/n are the Freundlich constants that represent adsorption capacity and adsorption intensity, respectively. Values of (1/n) < 1 imply chemisorption, while (1/n) > 1 indicate cooperative adsorption [34]. The maximum adsorption capacity of Cd^+2^ on the alumina spheres was 59.97 mg/g with its respective adsorption of 87 ppm of Cd^+2^.

The proposed adsorption mechanism is as follows. The alumina spheres have a negative surface charge at pH 5, so the electrostatic attractions between the material and the cation (Cd^+2^) are favored; subsequently, the formation of covalent bonds occurs between the oxygen of the alumina and the metal, as shown in Reaction (3). When the adsorption equilibrium is reached, a cooperative adsorption occurs where the adsorbate molecules form multilayers, causing the accumulation of cadmium on the material surface.
(R3)$$\equiv \mathrm{Al}-O^{-}+{\;\mathrm{Cd}}^{+2}\;\begin{array}{c} \overset{}{↔}\;\equiv \mathrm{Al}-O-\mathrm{Cd}-O-\mathrm{Al}\equiv \; \end{array}$$

Figure 16 shows SEM micrographs of the surface of alumina spheres treated at 1600 °C before and after adsorbing Cd(II). Figure 16C,D show the accumulation of what could be cadmium according to the Z contrast at the grain boundaries. Figure 17 shows the infrared spectrum of the alumina spheres before and after adsorbing cadmium. The spectrum shows the same bands corresponding to alumina described above with the addition of a broad, low-intensity band located around 1400 cm^−1^ corresponding to Cd-O bonds [35].

Table 5 summarizes the Cd^+2^ adsorbent capacity of various alumina-based materials (36–46), the point of zero charge of each material and its specific surface area. The α-Al_2_O_3_/Ba-β-Al_2_O_3_ spheres presented an adsorbent capacity of 59.97 mg/g at pH 5, which is higher than that of various materials and competes with materials that have a high surface area, which indicates that the surface charge is an important factor in the adsorbent capacity of heavy metals. In addition, the novelty of the material is its composition (α- Al_2_O_3_/Ba-β-Al_2_O_3_), since it has not been reported that Ba-β-Al_2_O_3_has heavy metal adsorbing capacity. Likewise, the scale of the material obtained at 4 mm in diameter makes it a manipulatable and easy-to-handle material. In addition to presenting high hardness, it makes it an applicable material to obtain filters, it is easy to regenerate since it can be given an acid treatment for the release of Cd(II), and it is friendly to the environment since it is an inert and non-toxic material, which makes it an applicable material for the removal of cadmium in aqueous media.

## 4. Conclusions

The synthesis of Ba-β-Al_2_O_3_ spheres was successfully obtained by the proposed synthesis route. The structure observed for Ba-β-Al_2_O_3_ was of the spinel type. The synthesis of Ba-β-Al_2_O_3_ is described in four main steps, (1) diffusion of barium and ion exchange, (2) release of barium by elimination of organic matter and breaking of the α-Al_2_O_3_ network, (3) oxidation and formation of BaO∙Al_2_O_3_ and (4) structural rearrangement and formation of Ba-β-Al_2_O_3_.

According to the pH vs. zeta potential curve, the spheres have an acid character and a negative surface charge causing the potential of−30 mV at pH 5, which favors electrostatic attractions between the material and the cation. Through adsorption studies, a Cd(II) adsorbent capacity of 59.97 mg/g (87 ppm of Cd(II)) was determined at pH 5 and 25 °C, according to the pseudo first and pseudo second models. The data describe the adsorption rate and depend on the available sites in the adsorbent for the physisorption and chemisorption process, and the Freundlich model indicates a cooperative adsorption. According to the composition, scale, properties and adsorbent capacity, the material obtained can be used as an adsorbent material for heavy metals in water filters.

## Figures and Tables

**Figure 1 materials-15-06809-f001:**
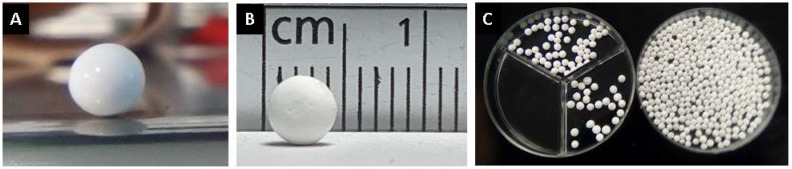
Photographs of alumina spheres: (**A**) without thermal treatment, (**B**) with treatment at 1600 °C and (**C**) several spheres at 1600 °C in petri dish.

**Figure 2 materials-15-06809-f002:**
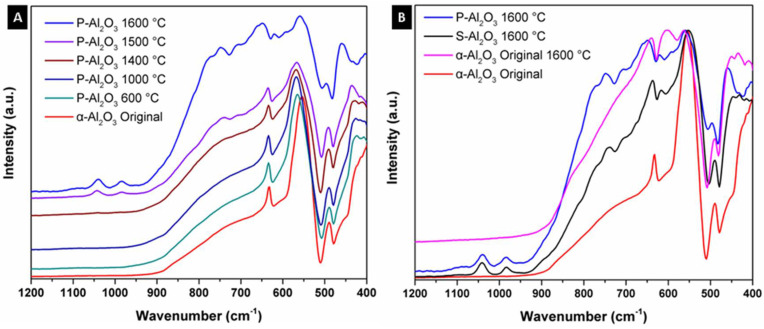
Infrared spectra of the alumina powders and sphere treated at different temperatures: (**A**) powdered sample (P-Al_2_O_3_) and (**B**) sphere (S-Al_2_O_3_).

**Figure 3 materials-15-06809-f003:**
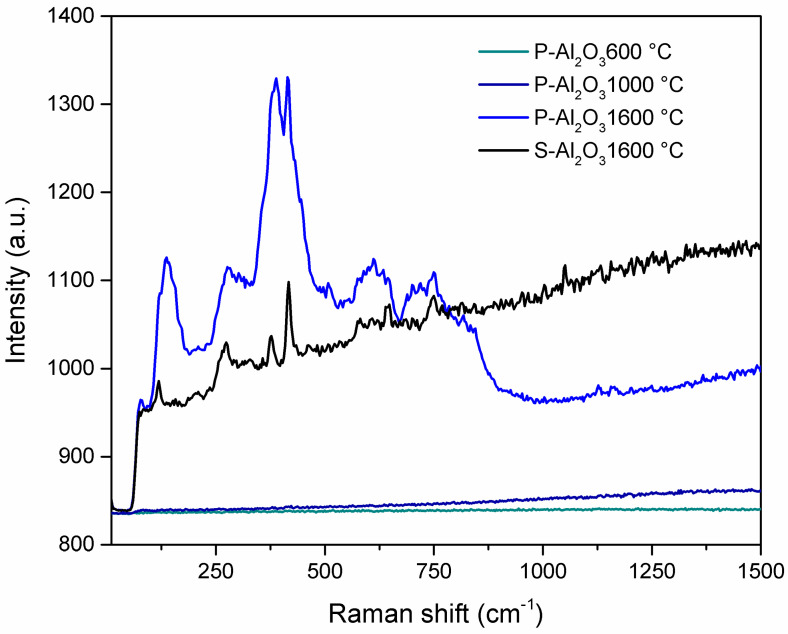
Raman spectra of the alumina powders and sphere treated at different temperatures.

**Figure 4 materials-15-06809-f004:**
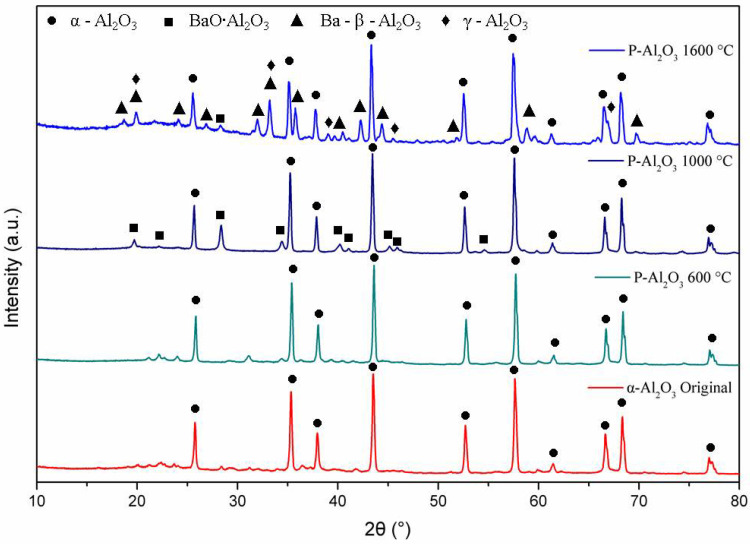
DRX of the alumina powders and sphere treated at different temperatures.

**Figure 5 materials-15-06809-f005:**
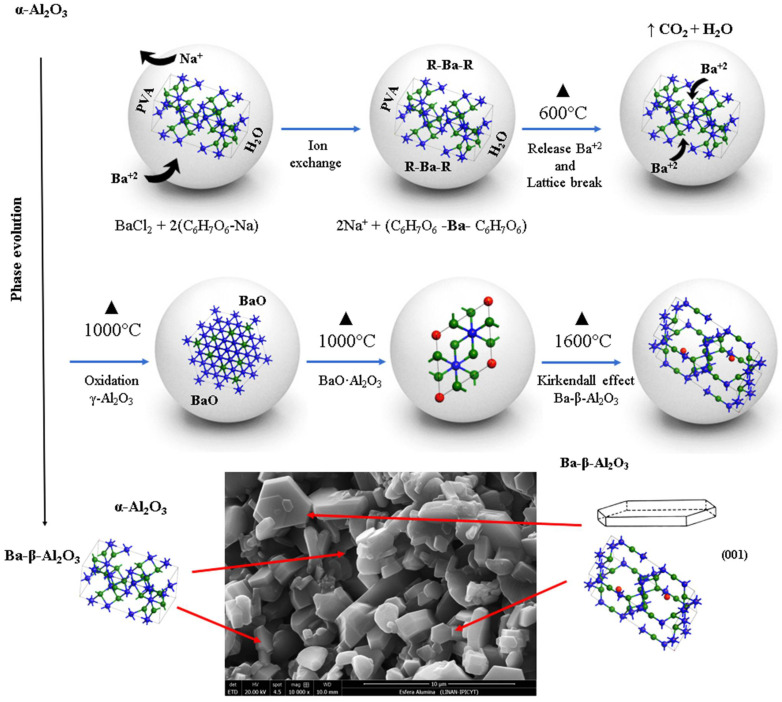
Scheme of synthesis mechanism of spinel Ba-β-Al_2_O_3_ in the spheres.

**Figure 6 materials-15-06809-f006:**
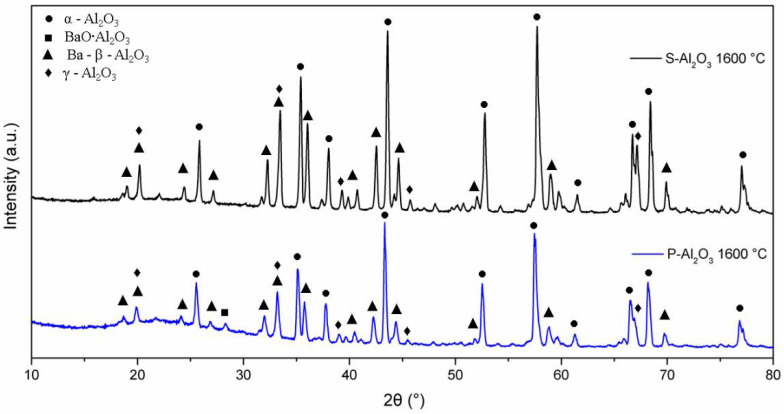
XRD diffractogram of powders and alumina sphere at 1600 °C.

**Figure 7 materials-15-06809-f007:**
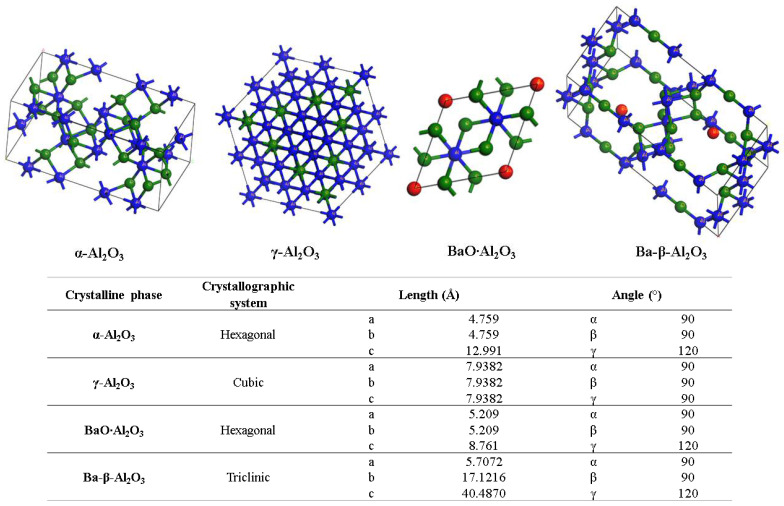
Crystalline systems and network parameters of the analyzed phases.

**Figure 8 materials-15-06809-f008:**
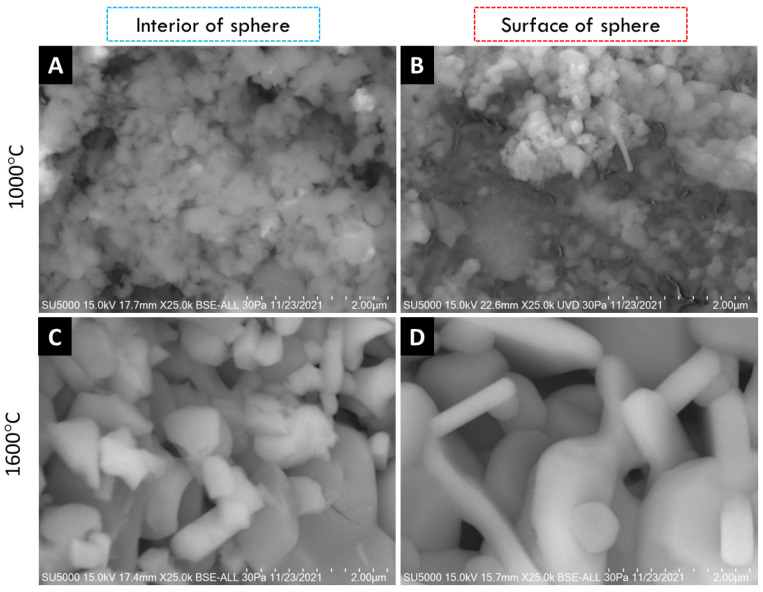
SEM micrographs of alumina spheres treated at 1000 °C (**A**,**B**) and 1600 °C (**C**,**D**).

**Figure 9 materials-15-06809-f009:**
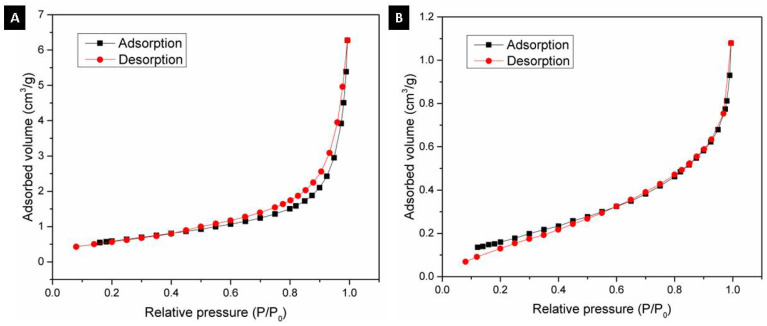
N_2_ adsorption–desorption isotherm: (**A**) Alumina sphere treated at 1000 °C and (**B**) Alumina sphere at 1600 °C.

**Figure 10 materials-15-06809-f010:**
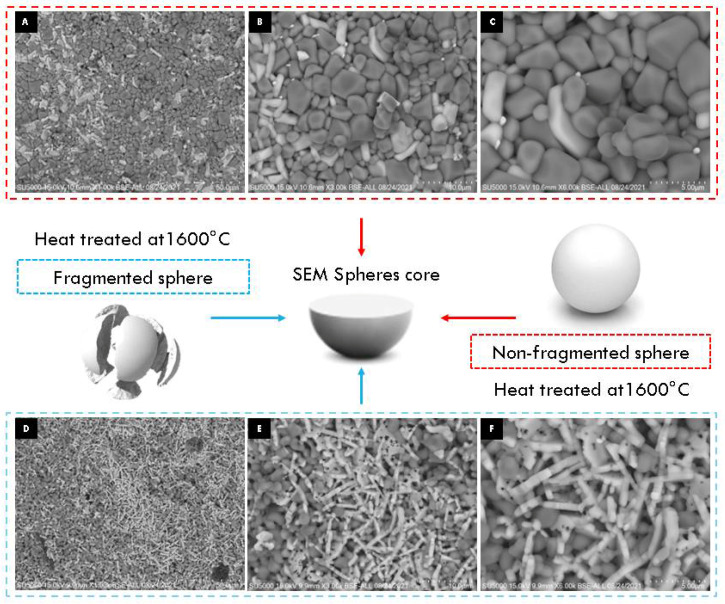
SEM micrographs of non-fragmented (**A**–**C**) and fragmented (**D**–**F**) alumina spheres treated at 1600 °C.

**Figure 11 materials-15-06809-f011:**
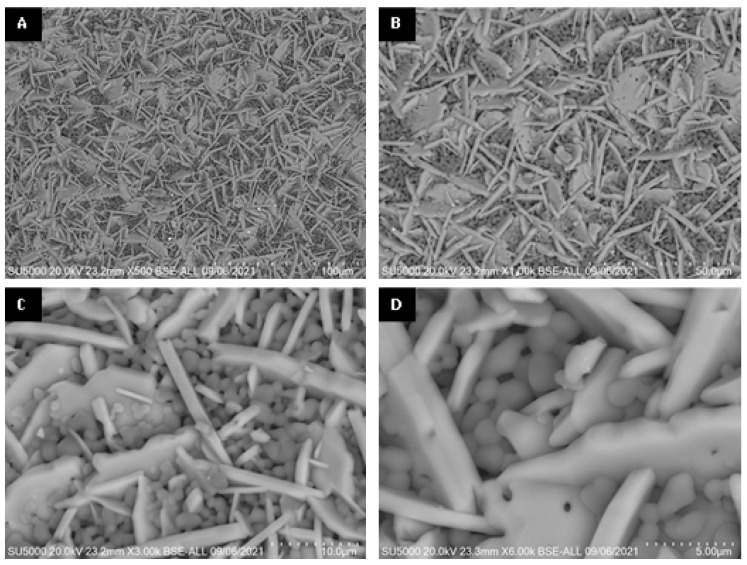
SEM micrographs of surface of alumina spheres treated at 1600 °C: (**A**) 500×, (**B**) 1000×, (**C**) 3000× and (**D**) 6000×.

**Figure 12 materials-15-06809-f012:**
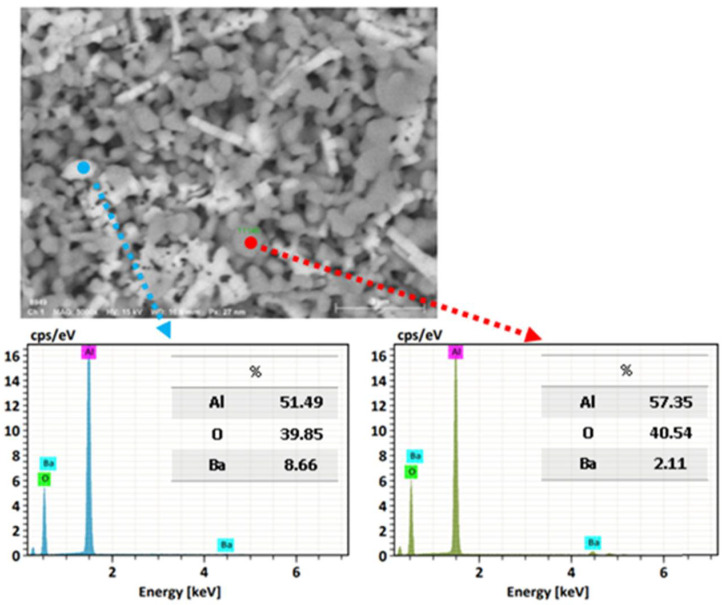
Microelemental surface analysis of alumina spheres treated at 1600 °C.

**Figure 13 materials-15-06809-f013:**
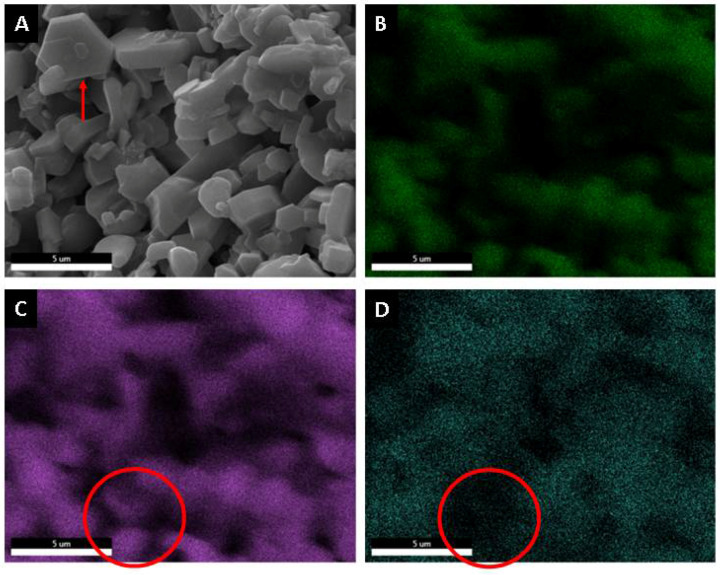
Microelemental mapping analysis surface of alumina spheres treated at 1600 °C. (**A**) Micrography, (**B**) Oxygen, (**C**) Aluminum and (**D**) Barium.

**Figure 14 materials-15-06809-f014:**
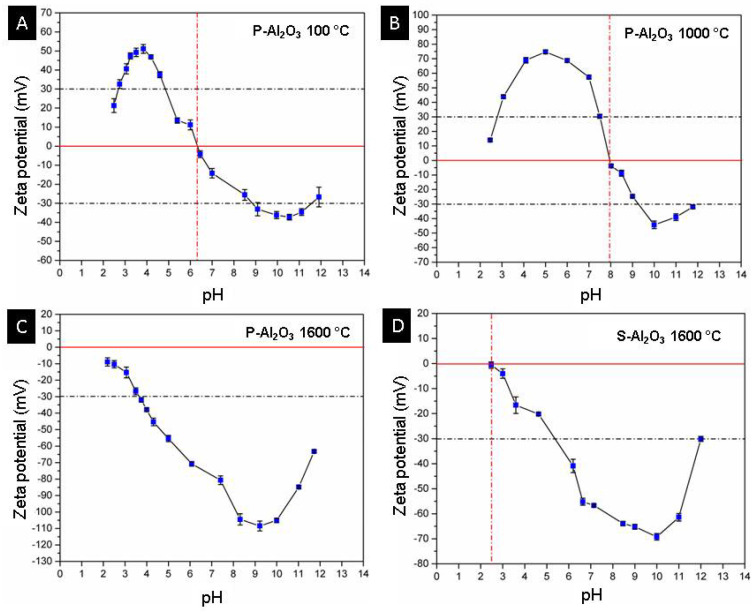
Curves of pH vs. zeta potential of powders and alumina sphere. (**A**) P-Al_2_O_3_ at 100 °C, (**B**) P-Al_2_O_3_ at 1000 °C, (**C**) P-Al_2_O_3_ 1600 °C and (**D**) S-Al_2_O_3_ at 1600 °C.

**Figure 15 materials-15-06809-f015:**
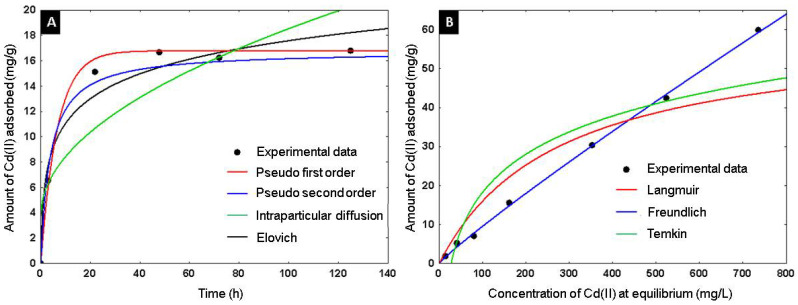
Adsorption of Cd(II) on alumina spheres at 1600 °C: (**A**) Adsorption kinetic models, (**B**) Adsorption equilibrium isotherm.

**Figure 16 materials-15-06809-f016:**
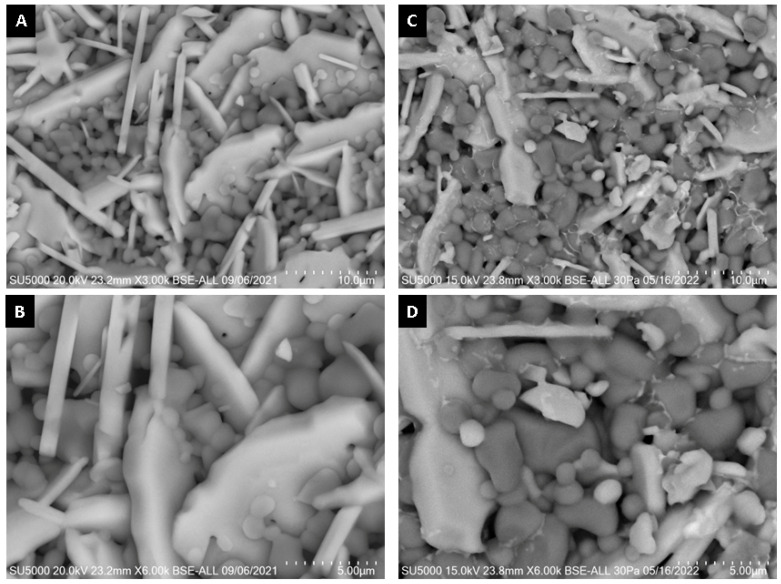
SEM surface micrographs of alumina spheres (**A**,**B**) before adsorption and (**C**,**D**) after Cd(II) adsorption.

**Figure 17 materials-15-06809-f017:**
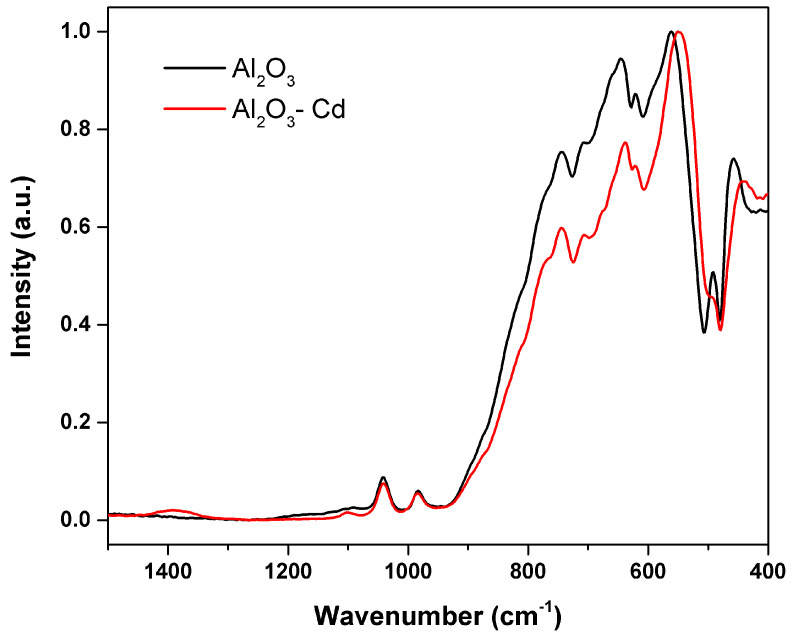
Infrared spectrum of S-Al_2_O_3_ 1600 °C and spheres after adsorbing cadmium.

**Table 1 materials-15-06809-t001:** Phase quantification for powders and alumina sphere.

Sample	(%) of Phase			
	α-Al_2_O_3_	γ-Al_2_O_3_	BaO∙Al_2_O_3_	Ba-β-Al_2_O_3_
P-Al_2_O_3_ 600 °C	92.67	7.21	0.12	-
P-Al_2_O_3_ 1000 °C	93.66	2.76	2.45	1.13
P-Al_2_O_3_ 1600 °C	93.66	3.92	0.84	1.58
S-Al_2_O_3_ 1600 °C	96.61	3.14	0.19	0.05

**Table 2 materials-15-06809-t002:** Textural properties of alumina spheres treated at 1000 and 1600 °C.

Material	Surface Area (m^2^/g)	Pore Volume (cm^3^/g)	Pore Size (nm)
S-Al_2_O_3_ 1000 °C	2.23	0.009707	17.43
S-Al_2_O_3_ 1600 °C	0.66	0.001669	10.19

**Table 3 materials-15-06809-t003:** Calculated kinetic parameters of Cd(II) adsorption on Al_2_O_3_ spheres.

Pseudo first order $q_{t}=q_{e}\left(1-e^{-\left(k_{1}t\right)}\right)$	*K*_1_ (min^−1^)	0.149
	*q_e_*	16.79
	%*D*	4.90
	*R* ^2^	0.993
Pseudo second order $q_{t}=\frac{k_{2}q_{e}{}^{2\;}t}{1+k_{2}q_{e}\;t\;}$	*K*_2_ (g mg^−1^ h^−1^)	0.015
	*q_e_*	16.29
	%*D*	4.67
	*R* ^2^	0.987
Elovich $q_{t}=\beta In\left(\alpha \beta t\right)$	*α* (mg g^−1^min)	30.65
	*β* (g mg^−1^)	0.15
	%*D*	8.63
	*R* ^2^	0.965
Intraparticular diffusion $q_{t}=K_{DI}t^{1/2}$	*K_D_* (mg/g/min)	1.47
	*I*	3.74
	%*D*	12.09
	*R* ^2^	0.784

**Table 4 materials-15-06809-t004:** Cd(II) adsorption isotherm parameters on alumina spheres at 1600 °C.

Langmuir $q_{e}=\frac{q_{m}bC_{e}}{1\;+\;bC_{e}}$	*b* (L/mg)	0.0036
	*q_m_*	43.60
	%*D*	36.17
	*R* ^2^	0.867
Freundlich $q_{e}=K_{F}C_{e}{}^{\frac{1}{n}}$	*K_F_* (mg/g)	0.1404
	1/*n*	1.092
	%*D*	3.83
	*R* ^2^	0.998
Temkin $q_{e}=\frac{RT}{b_{T}}In\;\left(K_{T}C_{e}\right)$	*K_T_* (L/g)	0.0363
	*b*	76.13
	%*D*	43.75
	*R* ^2^	0.836

**Table 5 materials-15-06809-t005:** Properties and adsorption capacities of Cd(II) of different alumina-based adsorbents.

References	Material/Form	Adsorbent Capacity (mg/g)	Point of Zero Charge (PZC)	Specific Surface Area (m^2^/g)
Present study	α-Al_2_O_3_/Ba-β-Al_2_O_3_/Spheres of 4 mm	59.97	2.50	0.66
[36]	Activated aluminum oxide (Al_2_O_3_)/Powder	35.06	6.51	126.00
[37]	Al_2_O_3_/MWCNTs/Powder	27.21	-	109.82
[38]	γ-Al_2_O_3_ (Ox-Al 7–80)/Powder	8.24	-	273.00
[39]	3DOM γ-alumina/Powder	23.32	6.40	77.30
[40]	Boehmite (γ-AlOOH)	6.39	8.60	246.40
[41]	OP (orange peles)-Al_2_O_3_ nanoparticles/Particles of 1 mm	19.12	4.06	-
[42]	glycerol-modified alumina nanoparticles/Powder	0.42	6.21	28.74
[43]	Mesoporous γ-Al_2_O_3_ synthesized from HAFA/Powder	88.26	6.62	318.68
[44]	PEI- Al_2_O_3_/micrometric hollow spheres	95.6	6.20	-
[45]	alumina/humic acid/Powder	13.9	7.00	-
[37]	Al_2_O_3_/MWCNTs/Powder	27.21	6.2	109.82

## Data Availability

The data presented in this study are available on request from the corresponding author.

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
