# Peer review of "Synthesis and Characterization of α-Al2O3/Ba-β-Al2O3 Spheres for Cadmium Ions Removal from Aqueous Solutions"

_materials, 2022, doi:10.3390/ma15196809_

Round 1
Reviewer 1 Report
The manuscript (materials-1862254) reports the results associated with synthesizing and characterizing several adsorbents based on nano-sized alumina. Finally, the α-Al2O3/Ba-β-Al2O3 has been applied to adsorb the Cd ions. The kinetic and isotherm models have also been performed to model the adsorption process.
After carefully reading this manuscript, I can confirm that it marginally passes the review phase. Although I really doubted rejecting this manuscript or suggesting a major revision, I finally decided to give the authors a golden chance to modify their manuscript. My comments are:
General comments:
1. The literature review is a mandatory part of the introduction section. Your manuscript has ignored this important part. You should review the recent articles associated with your study's main theme.
2. Did you correctly cite reference [1] in line 46?
3. Remove the dot symbol in the middle of line 58.
4. It is better not to use a bulk reference (like [1-3], [5-7], and so on) in the text. It would help if you placed each reference in its correct location. In this way, the readers can easily refer to a valid reference for your claimed statements.
5. Add a paragraph to the end of the introduction and highlight the contribution of your study in the investigated field and its novelty.
6. Several statements in the introduction need valid references. For example, the last part of line 60 can be justified using this reference (https://doi.org/10.1016/j.eti.2021.101439).
7. Break down the "Results and discussion" section into several meaningful subsections.
8. Generally, all the texts added to the figures (specifically the adsorbent names) must be modified to become easily readable.
9. Improve the quality of Figure 4.
1. Y-title of Figures 13A, 13B, 13C, and 13D has a typo (I think it may be written in a language other than English).
1. The legends, y-title, and x-title of Figures 14A and 14B are not in English!!!
1. All equations should be numbered. Present the R2 formula.
1. Present the mathematical formula of kinetic and equilibrium models utilized in Tables 3 and 4.
1. I think you should pay more attention to reporting results for the Cd removal in both the abstract and conclusion sections. You may report the best adsorbent, the highest removal capacity, the optimum operating conditions that maximize the Cd removal, the most suitable isotherm and kinetic models, and so on.
1. Line 396: stars or starts?
1. Rewrite/simplify the first four-line long sentence of the Conclusions.
1. Follow the same structure for preparing the Reference list. Furthermore, some of the references are not completely introduced (either their volume, publication year, page, ... were missed)
Scientific comments:
1. I don't know why you highlighted the "spherical adsorbent" in your manuscript. The adsorbent should be readily available, reusable, and has high removal capacity.
2. The authors should introduce the general information of a solution that its Cd ions have been removed. Did you work with an actual or synthetic solution or both?
3. The pseudo-second-order kinetic model has two adjustable coefficients (). Why did you only report the K2 value? The Langmuir isotherm has two adjustable coefficients. Surprisingly, you only reported one of them. This paper may help you (https://doi.org/10.1080/19443994.2015.1091748). Furthermore, please check all other kinetic/equilibrium models to resolve such a mistake.
4. It is necessary to check the reusability of your synthesized adsorbent.
55. The specific surface area of your synthesized material (Table 5) is too small. Since the surface area is a key feature of adsorbent, how do you justify this limitation?
Author Response
- 1. The literature review is a mandatory part of the introduction section. Your manuscript has ignored this important part. You should review the recent articles associated with your study's main theme.
Response: corrected in manuscript
- Did you correctly cite reference [1] in line 46?
Response: corrected reference in manuscript
- Remove the dot symbol in the middle of line 58.
Response: corrected in manuscript
- It is better not to use a bulk reference (like [1-3], [5-7], and so on) in the text. It would help if you placed each reference in its correct location. In this way, the readers can easily refer to a valid reference for your claimed statements.
Response: corrected in manuscript
- Add a paragraph to the end of the introduction and highlight the contribution of your study in the investigated field and its novelty.
Response: added in manuscript
- Several statements in the introduction need valid references. For example, the last part of line 60 can be justified using this reference (https://doi.org/10.1016/j.eti.2021.101439).
Response: corrected in manuscript
- Break down the "Results and discussion" section into several meaningful subsections.
Response: corrected in manuscript
- Generally, all the texts added to the figures (specifically the adsorbent names) must be modified to become easily readable.
Response: corrected in manuscript
- Improve the quality of Figure 4.
Response: corrected in manuscript
- Y-title of Figures 13A, 13B, 13C, and 13D has a typo (I think it may be written in a language other than English).
Response: corrected in manuscript
- The legends, y-title, and x-title of Figures 14A and 14B are not in English!!!
Response: corrected in manuscript
- All equations should be numbered. Present the R2 formula.
Response: corrected and added in manuscript
- Present the mathematical formula of kinetic and equilibrium models utilized in Tables 3 and 4.
Response: added in manuscript
- I think you should pay more attention to reporting results for the Cd removal in both the abstract and conclusion sections. You may report the best adsorbent, the highest removal capacity, the optimum operating conditions that maximize the Cd removal, the most suitable isotherm and kinetic models, and so on.
Response: corrected in manuscript
- Line 396: stars or starts?
Response: corrected in manuscript
- Rewrite/simplify the first four-line long sentence of the Conclusions.
Response: corrected in manuscript
- Follow the same structure for preparing the Reference list. Furthermore, some of the references are not completely introduced (either their volume, publication year, page, ... were missed)
Response: corrected in manuscript
Scientific comments:
- I don't know why you highlighted the "spherical adsorbent" in your manuscript. The adsorbent should be readily available, reusable, and has high removal capacity.
Response: corrected in manuscript
- The authors should introduce the general information of a solution that its Cd ions have been removed. Did you work with an actual or synthetic solution or both?
Response: corrected in manuscript
- The pseudo-second-order kinetic model has two adjustable coefficients (). Why did you only report the K2 value? The Langmuir isotherm has two adjustable coefficients. Surprisingly, you only reported one of them. This paper may help you (https://doi.org/10.1080/19443994.2015.1091748). Furthermore, please check all other kinetic/equilibrium models to resolve such a mistake.
Response: corrected in manuscript
- It is necessary to check the reusability of your synthesized adsorbent.
Response: According to the curve of pH vs zeta potential, at acid pH there is only a change in the potential of the adsorbent material, so it can be reused by giving it an acid treatment to release the cadmium ions. Corrected in manuscript
- The specific surface area of your synthesized material (Table 5) is too small. Since the surface area is a key feature of adsorbent, how do you justify this limitation?
Response: Due to the sintering process, the surface area is reduced, however, this same process helps to consolidate the material and obtain a manipulable material with high hardness, which is applicable for obtaining filters. Although it has a low surface area, it has other properties of a good absorbent material, in this case the main property is the surface charge, the more negative the surface charge of the material, the greater the adsorbent capacity of cations (heavy metals) due to to the electrostatic attractions that exist between both components.

Reviewer 2 Report
This manuscript addresses a very interesting topic. However, There is a certain differentiation between the research purpose and the content of the manuscript . It says that “Ceramics materials at low temperatures has been studied” in the introduction, but it is not reflected in the manuscript. The adsorption capacity of the adsorbent material is not very high compared to the data in Ref. Besides,the results are not enough. I think this manuscript can accept after minor revision.
Author Response
Response: corrected in manuscript
Reviewer 3 Report
Review of paper ‘Synthesis and characterization of α-Al2O3/Ba-β-Al2O3 spheres for Cadmium Ions Removal from Aqueous Solutions’ prepared by Pamela Nair Silva-Holguín, Álvaro de Jesús Ruíz-Baltazar, Nahum Medellín-Castillo, Gladis Judith Labrada Delgado, and Simón Yobanny Reyes-López.
Manuscript materials- -1862254 is focused on the presentation of new adsorbent based on aluminum for the removal of cadmium(II) ions. The method of preparation of sorbent as well as adsorption study and mechanism proposition have been presented. I have some suggestions that authors may consider prior to publication of this work:
1. The authors should explain the choice of concentrations of cadmium ions and pH during the experiments. Was this dictated by the concentration values that may be present in the wastewater?
2. The authors in this paper carried out a number of tests to characterise the materials studied, for example FTIR, SEM, XRD. Of course, the characterisation itself is important, but it is also worth adding information on how the parameters determined affect possible sorption. In this way, desired directions for synthesis could be indicated.
3. The authors should compare their own results with those of other adsorbents, not only alumina-based adsorbents, for cadmium(II) ions removal. Only in this way is it possible to indicate the added value for the new material. Without such a comparison, the innovation and novelty of the own solution cannot be confirmed.
4. The paper lacks any mention of the possibility of regenerating the adsorbent, reusing it, or managing it after the process. From a practical point of view, this information should be added in the paper.
5. The reactions should be corrected:
- Reaction 1: no balance for chlorine and charge,
- Reaction 2: does not balance, four bonds for aluminium (?),
- Reaction 3: does not balance, four bonds for aluminium (?).
6. Some figures need editing correction. In their current form Figures 6, 7 and 15 are blurred. In addition, in Figure 14, the description of the axes should be changed to English.
7. The work requires editing and language correction. Some examples below:
- lines 57-58: 2 dots are unnecessary in a sentence,
- in equation (1), the subscripts should be added so that the symbols are consistent with the description below,
- one temperature notation should be used in the work, currently authors sometimes use spaces and sometimes not, see for example ‘At 600°C’ in line 197 and ‘At 1000 °C’ in line 200,
- according to IUPAC (see Brief Guide to the Nomenclature of Inorganic Chemistry) notation of ions should be without space Cd(II) not Cd (II).
- English correction: line 17 ‘is performed with structural’, line 45 ‘the sequence of transition of alumina that are is’.
Author Response
Manuscript materials- -1862254 is focused on the presentation of new adsorbent based on aluminum for the removal of cadmium(II) ions. The method of preparation of sorbent as well as adsorption study and mechanism proposition have been presented. I have some suggestions that authors may consider prior to publication of this work:
1.The authors should explain the choice of concentrations of cadmium ions and pH during the experiments. Was this dictated by the concentration values that may be present in the wastewater?
Response: The pH was chosen because, according to the cadmium speciation diagram, we find it as a divalent cation, and it is the species that we want to adsorb. Regarding the concentrations, a wide range of 20 to 800 mg/L of Cd(II) was chosen to be able to observe the behavior of the material in its adsorbent capacity at different initial concentrations.
- The authors in this paper carried out a number of tests to characterise the materials studied, for example FTIR, SEM, XRD. Of course, the characterisation itself is important, but it is also worth adding information on how the parameters determined affect possible sorption. In this way, desired directions for synthesis could be indicated.
Response: corrected in manuscript
- The authors should compare their own results with those of other adsorbents, not only alumina-based adsorbents, for cadmium(II) ions removal. Only in this way is it possible to indicate the added value for the new material. Without such a comparison, the innovation and novelty of the own solution cannot be confirmed.
Response: The purpose of comparing the adsorbent capacity of the material obtained with various alumina-based materials is to be able to see that the surface charge is an important factor in the adsorbent capacity of heavy metals and that a high surface area is not necessary. In addition to introducing β-alumina as an adsorbent material, since γ-alumina is mainly used in adsorption processes. Added reference [46].
- The paper lacks any mention of the possibility of regenerating the adsorbent, reusing it, or managing it after the process. From a practical point of view, this information should be added in the paper.
Response: According to the curve of pH vs zeta potential, at acid pH there is only a change in the potential of the adsorbent material, so it can be reused by giving it an acid treatment to release the cadmium ions.
- The reactions should be corrected:
- Reaction 1: no balance for chlorine and charge,
Response: corrected in manuscript
- Reaction 2: does not balance, four bonds for aluminium (?),
Response: el símbolo ≡ corresponde a la superficie del material
- Reaction 3: does not balance, four bonds for aluminium (?).
Response: el símbolo ≡ corresponde a la superficie del material
- Some figures need editing correction. In their current form Figures 6, 7 and 15 are blurred. In addition, in Figure 14, the description of the axes should be changed to English.
Response: corrected in manuscript
- The work requires editing and language correction. Some examples below:
- lines 57-58: 2 dots are unnecessary in a sentence,
- in equation (1), the subscripts should be added so that the symbols are consistent with the description below,
Response: corrected in manuscript
- one temperature notation should be used in the work, currently authors sometimes use spaces and sometimes not, see for example ‘At 600°C’ in line 197 and ‘At 1000 °C’ in line 200,
Response: corrected in manuscript
- according to IUPAC (see Brief Guide to the Nomenclature of Inorganic Chemistry) notation of ions should be without space Cd(II) not Cd (II).
Response: corrected in manuscript
- English correction: line 17 ‘is performed with structural’, line 45 ‘the sequence of transition of alumina that are is’.
Response: corrected in manuscript
Round 2
Reviewer 1 Report
Dear authors
Your response to the reviewer file should guide referees to easily find and check your modifications in the manuscript by introducing page and line numbers (for example, this comment has been addressed in page XX, Line YY of the revised manuscript).
It is not a good idea to only respond to the referees' comments by "corrected in manuscript".
I'm looking forward to receiving your revised manuscript in the soon future.
Regards,
Author Response
Thank you very much for your comments, the corrections were marked in red and green in the submitted manuscript, an apology if any inconvenience was caused.
Round 3
Reviewer 1 Report
Dear authors,
Do you think I did not check your revised manuscript, and I did not see that you highlighted it in red and green colors?!!!!
As the last chance, please clearly guide me to find your modification related to each of my comments as follows:
Comment 1: ....
Author's response: The modification has been made on Page XX1, Line YY1 of the revised manuscript.
Comment 2: ...
Author's response: the modification has been made on Page XX2, Line YY2 of the revised manuscript.
and so on. You should follow this procedure till reach the last comment.
If your new version of the "response to reviewer's comments" file does not prepare based on this suggestion, I have to suggest a rejection for your manuscript.
Regards,
Author Response
- The literature review is a mandatory part of the introduction section. Your manuscript has ignored this important part. You should review the recent articles associated with your study's main theme.
Response: Added references on page 2, lines 51 and 67-68.
- Did you correctly cite reference [1] in line 46?
Response: The citation was corrected in the references section, page 19, line 530-531.
- Remove the dot symbol in the middle of line 58.
Response: the symbol was removed from the line 58.
- It is better not to use a bulk reference (like [1-3], [5-7], and so on) in the text. It would help if you placed each reference in its correct location. In this way, the readers can easily refer to a valid reference for your claimed statements.
Response: Added and separated references on page 2, lines 51 and 67-68.
- Add a paragraph to the end of the introduction and highlight the contribution of your study in the investigated field and its novelty.
Response: added a paragraph on page 3, lines 96 to 103.
- Several statements in the introduction need valid references. For example, the last part of line 60 can be justified using this reference (https://doi.org/10.1016/j.eti.2021.101439).
Response: Added references on page 2, lines 51 and 67 to 68.
- Break down the "Results and discussion" section into several meaningful subsections.
Response: Added subsections on page 5 line 197, page 14 line 380, and page 15 line 418.
- Generally, all the texts added to the figures (specifically the adsorbent names) must be modified to become easily readable.
Response: the name of the figures of the pages was modified: page 5 line 223-224, page 6 line 234-235, page 7 line 272, page 8 line 293-294, page 9 line 297, page 10 line 318, page 12 line 363-364, page 14 line 414-415, page 15 line 454, page 16 line 457 and 460, page 17 line 481 and 484.
- Improve the quality of Figure 4.
Response: improved the quality of figure 4 on page 8 line 290 (new image is added)
- Y-title of Figures 13A, 13B, 13C, and 13D has a typo (I think it may be written in a language other than English).
Response: corrected language in figure 13 on page 14 line 413
- The legends, y-title, and x-title of Figures 14A and 14B are not in English!!!
Response: corrected language in figure 14 on page 15 line 453
- All equations should be numbered. Present the R2 formula.
Response: equations are numbered: page 4 lines 150, 152, 154, 156, 169, 170, 171, 182, 189 and page 15 line 236. Added formula R2 on page 4 line 189
- Present the mathematical formula of kinetic and equilibrium models utilized in Tables 3 and 4.
Response: the equations were added on page 4 lines 150, 152, 154, 156, 169, 170, 171,
- I think you should pay more attention to reporting results for the Cd removal in both the abstract and conclusion sections. You may report the best adsorbent, the highest removal capacity, the optimum operating conditions that maximize the Cd removal, the most suitable isotherm and kinetic models, and so on.
Response: the modifications were made on page 1 line 28-31 and on page 18 line 511-513 and page 19 line 514-517
- Line 396: stars or starts?
Response: starts, conclusions were modified on page 18 and 19 lines 503-517
- Rewrite/simplify the first four-line long sentence of the Conclusions.
Response: conclusions were modified on page 18 and 19 lines 503-517
- Follow the same structure for preparing the Reference list. Furthermore, some of the references are not completely introduced (either their volume, publication year, page, ... were missed)
Response: references are restructured, completed and references are added on pages 19-21 lines 535-633
Scientific comments:
- I don't know why you highlighted the "spherical adsorbent" in your manuscript. The adsorbent should be readily available, reusable, and has high removal capacity.
Response: corrected on page 1 line 16
- The authors should introduce the general information of a solution that its Cd ions have been removed. Did you work with an actual or synthetic solution or both?
Response: the information was added on page 3 line 140-143.
- The pseudo-second-order kinetic model has two adjustable coefficients (). Why did you only report the K2 value? The Langmuir isotherm has two adjustable coefficients. Surprisingly, you only reported one of them. This paper may help you (https://doi.org/10.1080/19443994.2015.1091748). Furthermore, please check all other kinetic/equilibrium models to resolve such a mistake.
Response: the information was added on table # 3 and # 4 on page 16 line 458 and 461.
- It is necessary to check the reusability of your synthesized adsorbent.
Response: the information was added on page 15 lines 416-421.
- The specific surface area of your synthesized material (Table 5) is too small. Since the surface area is a key feature of adsorbent, how do you justify this limitation?
Response: Due to the sintering process, the surface area is reduced, however, this same process helps to consolidate the material and obtain a manipulable material with high hardness, which is applicable for obtaining filters. Although it has a low surface area, it has other properties of a good absorbent material, in this case the main property is the surface charge, the more negative the surface charge of the material, the greater the adsorbent capacity of cations (heavy metals) due to to the electrostatic attractions that exist between both components.

Round 4
Reviewer 1 Report
Dear authors,
Thank you very much for thoroughly addressing my comments, even after several rounds of revision.
Your manuscript is now acceptable for publication in the Materials journal.
Regards,
Author Response
Thank you very much for the comments,
Regards,